# Multifaceted secretion of htNSC-derived hypothalamic islets induces survival and antidiabetic effect via peripheral implantation in mice

Yizhe Tang[†], Juan Pablo Zuniga-Hertz[†], Cheng Han, Bin Yu, Dongsheng Cai*

Department of Molecular Pharmacology, Diabetes Research Center, Institute for Aging Research, Albert Einstein College of Medicine, Bronx, United States

**Abstract** We report that mouse hypothalamic stem/progenitor cells produce multiple pancreatic, gastrointestinal and hypothalamic peptides in addition to exosomes. Through cell sorting and selection according to insulin promoter activity, we generated a subpopulation(s) of these cells which formed 3D spherical structure with combined features of hypothalamic neurospheres and pancreatic islets. Through testing streptozotocin-induced pancreatic islet disruption and fatal diabetes, we found that peripheral implantation of these spheres in mice led to remarkable improvements in general health and survival in addition to a moderate antidiabetic effect, and notably these pro-survival versus metabolic effects were dissociable to a significant extent. Mechanistically, secretion of exosomes by these spheres was essential for enhancing survival while production of insulin was important for the antidiabetic effect. In summary, hypothalamic neural stem/progenitor cells comprise subpopulations with multifaceted secretion, and their derived hypothalamic islets can be implanted peripherally to enhance general health and survival together with an antidiabetic benefit.

**\*For correspondence:**
dongsheng.cai@einstein.yu.edu

[†]These authors contributed equally to this work

**Competing interests:** The authors declare that no competing interests exist.

## Introduction

Neural stem/progenitor cells (NSCs) are essential for brain development during the embryonic stage. On the other hand, some of NSCs continue to exist in a few brain regions of rodents including the sub-ventricular zone and the dentate gyrus which require adult neurogenesis (*Gage, 2000*; *Gross, 2000*; *Morrison, 2001*; *Alvarez-Buylla and Lim, 2004*; *Emsley et al., 2005*; *Opendak and Gould, 2015*). The hypothalamus of animals such as rodents was recently found to be another brain region with certain although limited neurogenic activity during the adulthood (*Kokoeva et al., 2005*; *Kokoeva et al., 2007*; *Pierce and Xu, 2010*; *Migaud et al., 2010*; *Pérez-Martín et al., 2010*; *Li et al., 2012*; *Lee et al., 2012*; *McNay et al., 2012*; *Wang et al., 2012*; *Robins et al., 2013*; *Sousa-Ferreira et al., 2014*; *Zhang et al., 2017*; *Recabal et al., 2017*; *Pellegrino et al., 2018*). We showed that the hypothalamus of adult mice contained Sox2-positive NSCs (referred to as htNSCs), residing predominantly in the hypothalamic third-ventricle wall and the mediobasal hypothalamus (MBH) parenchyma, which are capable of differentiation into three neural linages (*Li et al., 2012*). These cells can play a role in long-term metabolic control through the effects on hypothalamic neural plasticity, which could be suggested in the context of relevant literature (*Pierce and Xu, 2010*; *Li et al., 2012*; *Lee et al., 2012*; *McNay et al., 2012*; *Scarlett et al., 2016*). Also, some of these cells in the third-ventricle wall and median eminence which are known as tanycytes could influence metabolic physiology through the gate-keeping functions such as hormone transporting or affecting the blood-brain barrier (*Langlet, 2014*), but it remains unclear if these cells could have some endocrine action to directly regulate metabolic balance. Recently, we discovered that htNSCs have a key

endocrine function through secreting exosomes and contribute to the hypothalamic control of aging (*Zhang et al., 2017*). This research progress points to htNSCs as a cell therapy model based on the endocrine feature of these cells; thus, it is scientifically demanding to gain in-depth understandings into the endocrinology of these cells. Moreover, since delivery of these cells into the brain is a hurdle for clinical application, we decided to explore alternative application approaches such as peripheral implantation. In this work, we discovered that a subpopulation(s) of htNSCs can form spherical structure with combined features of secreting multiple pancreatic/gustatory and hypothalamic factors, and peripheral implantation of these spheres leads to substantial improvements in general health and survival together with a moderate anti-diabetic effect in a fatal diabetic model, while the underlying endocrine mechanisms were comprehensively attributed to multifaceted secretion of these spheres.

## Results

### Co-expression of multiple peptidyl hormones by some htNSCs

Using quantitative PCR, we found that cultured htNSCs expressed mRNAs of multiple hypothalamic and gastrointestinal hormones, evidently including pancreatic hormones such as insulin, glucagon, and somatostatin which led to our research interest of this study. In this background, we did immunostaining revealing that some subpopulations of htNSCs co-expressed insulin, glucagon and somatostatin (*Figure 1a,b*, *Figure 1—figure supplement 1*). All these peptides were found to exist in the vesicles within the cytoplasm of some htNSCs, according to immunostaining signals in puncta pattern revealed by these peptide antibodies. These peptides could be localized in the same or different vesicles; for instance, insulin-positive vesicles were often distinct from the vesicles positive for glucagon and somatostatin, while the latter two peptides were often co-localized in the same vesicles (*Figure 1a,b*). Using large dense-core vesicles (LDCV) and synaptic-like microvesicles (SLMV) markers (*Figure 1—figure supplement 2*), we revealed that insulin was present mostly in the LDCV while glucagon and somatostatin were often co-present predominantly in the SLMV.

We then employed genetic approaches to evaluate the transcriptional capabilities of these cells in producing pancreatic hormones. To do so, we transfected primary htNSCs with Flp recombinase-dependent GFP reporter plasmid whose expression is driven by insulin, glucagon or somatostatin promoter (namely *Ins*-Flp, *Gcg*-Flp and *Sst*-Flp, respectively). The *Ins* promoter activity was found in ~7% of htNSCs, while the *Gcg* and *Sst* promoter activities were found in ~2% and~3% of htNSCs, respectively (*Figure 1c*), indicating that only small subpopulations of these cells significantly shared the genetic features of pancreatic islet cells.

We also examined if different pancreatic peptides are produced in same or different htNSCs. To do so, we constructed a Flp/Cre dual fluorescence reporter system (*Figure 1—figure supplement 3a*), as this system allowed us to simultaneously detect the gene promoter activities of two different peptides in the same cells. The specificity of Flp- versus Cre-dependent fluorescence in this reporter system was evaluated in various cell models including an unrelated cell line HEK293T (*Figure 1—figure supplement 3b*). Then, htNSCs were transfected with this reporter system together with Flp and Cre plasmids driven by different promoters, as indicated in *Figure 1—figure supplement 4*. We found that among htNSCs which displayed *Ins* promoter activation,~25% of them showed co-activation of *Sst* promoter, and 20% of them showed co-activation of *Gcg* promoter. We also found that ~44% of htNSCs with *Gcg* promoter activation showed co-activation of *Sst* promoter. Thus, multiple peptidyl hormones are co-expressed in subpopulations of htNSCs. According to the literature (*Katsuta et al., 2010*; *Segerstolpe et al., 2016*; *Enge et al., 2017*; *De Krijger et al., 1992*; *van der Meulen and Huising, 2014*; *Jeon et al., 2009*), co-existence of these pancreatic hormones in same cells could also be seen in some immature islets of Langerhans, indicating that there exists some developmental relationship between these htNSCs and the precursors of pancreatic islets.

In addition to pancreatic peptides, we observed that many of these htNSCs expressed gastrointestinal hormones such as galanin, vasoactive intestinal polypeptide (VIP), cholecystokinin (CCK) and pituitary adenylate cyclase-activating peptide (PACAP) as well as hypothalamic peptides such as neuropeptide Y (NPY), agouti-related peptide (AgRP), alpha melanocyte-stimulating hormone (α-MSH), corticotrophin-releasing factor (CRF), gonadotropin-releasing hormone (GnRH) and oxytocin in subcellular vesicles (some examples of these peptide immunostainings are shown in *Figure 1—figure*

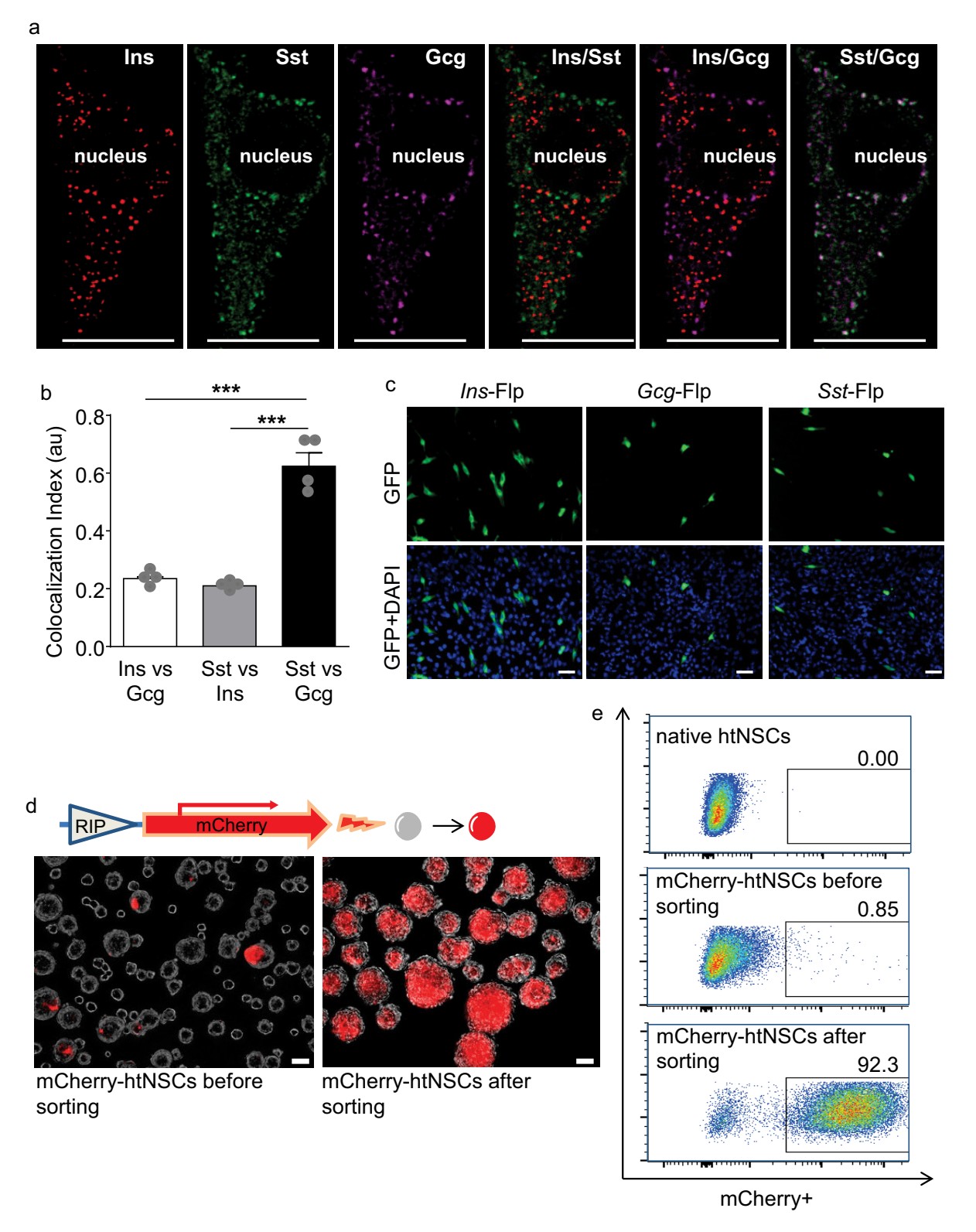

**Figure 1.** Some htNSCs expressing pancreatic hormones and cell selection via insulin promoter. (**a–b**) Representative immunostaining of single-cell confocal images. Cultured htNSCs were co-immunostained for somatostatin (Sst), insulin (Ins) and glucagon (Gcg); images were merged to show co-localization of these peptides in the same single cell (**a**) while co-localization index (au: arbitrary unit) of any two peptide types were calculated (**b**). (**c**) Cultured htNSCs were transfected with Flp-dependent GFP expressing plasmid together with Flp plasmid driven by insulin promoter (*Ins*-Flp),

*Figure 1 continued on next page*

*Figure 1 continued*

glucagon prompter (*Gcg*-Flp) and somatostatin promoter (*Sst*-Flp). (**d**) A line of htNSCs was virally infected to stably express RIP-driven mCherry and transiently infected by adenoviral cocktail of expressing Ngn3, Pdx-1 and MafA (d left panel) leading to the establishment of mCherry-positive subpopulation (d right panel). (**e**) FACS for the established subpopulation of mCherry-positive cells compared to native htNSCs and these cells after cocktail induction prior to sorting. Scale Bar: 10 µm (**a**) or 50 µm (**c, d**). ***p<0.001 (ANOVA and post-hoc test), n = 4 independent biological samples per group (**b**); data are mean ± s.e.m.

The online version of this article includes the following figure supplement(s) for figure 1:

**Figure supplement 1.** Insulin, glucagon and somatostatin staining of htNSCs.
**Figure supplement 2.** Vesicle markers for pancreatic peptidyl hormones in htNSCs.
**Figure supplement 3.** Cre/Flp dual fluorescence reporter system.
**Figure supplement 4.** Co-expression of pancreatic peptidyl hormones in htNSCs.
**Figure supplement 5.** Vesicle localization of gastrointestinal and hypothalamic peptides in htNSCs.

*supplement 5*). Notably, for any pairs of these peptides, they were always found to co-exist in some of htNSCs, and each of these peptides could be found present in insulin-positive htNSCs. Thus, htNSCs contain subpopulations that produce multiple peptides in bulk as well as single-cell levels.

## Generation of insulin promoter-selected htNSC spheres

Because insulin is a predominant pancreatic hormone, we designed to employ insulin promoter activity as a target for isolating a subpopulation(s) of htNSCs. To do so, we infected htNSCs with a lentiviral system containing mCherry expression driven by rat insulin promoter (RIP) which fluorescently labeled such population(s) of htNSCs. In our initial trial, indeed some htNSCs were made mCherry-positive by this method; however, most likely because RIP activity was not strong, it was insufficient for fluorescent flow cytometry (FACS) sorting-based cell isolation. To address this limitation, we increased the RIP activity through temporarily introducing Ngn3, Pdx-1 and MafA transcription factor cocktail to the cells, as suggested by the literature (*Zhou et al., 2008*; *Chen et al., 2014*; *Ariyachet et al., 2016*). After this cocktail was transiently applied to htNSCs via an adenoviral induction, RIP-driven mCherry fluorescence strikingly increased in many htNSCs (*Figure 1d*, left panel). With this success, mCherry-positive cells were abundantly sorted out via FACS (*Figure 1e* top vs. middle panel) and were subsequently established as a subpopulation(s) of insulin promoter-selected cells, as evaluated with microscopy (*Figure 1d* right panel) and also FACS (*Figure 1e* bottom panel). After a few generations of cell passaging, through which the cocktail completely disappeared, we confirmed that RIP-driven mCherry remained strong in a majority of cells indicating that these cells were induced to persistently have an enhanced level of insulin promoter activity.

## Multifaceted peptide secretion of insulin promoter-selected htNSC spheres

We assessed if these insulin promoter-selected htNSCs were able to release insulin. First, using a luminescent reporter which was known to sensitively report insulin secretion (*Burns et al., 2015*), we found that insulin secretion substantially increased in these insulin promoter-selected htNSCs compared to insulin secretion from htNSCs of pre-selection (*Figure 2a*). Using an ELISA which directly measured insulin, we found that these insulin promoter-selected htNSCs secreted insulin in a glucose-regulated manner. As shown in *Figure 2b*, we designed a test in which these htNSCs were subjected to 3 cycles of low-level (2 mM) and high-level (20 mM) glucose incubation, 1 hr each step. We found that high-level glucose led to ~3 fold increase of insulin release, which dropped to the baseline after glucose returned to the low level. Subsequent two repeats of high-glucose stimulation were still able to increase insulin release but in a gradually declining manner. Moreover, we subjected these htNSCs to potassium chloride or arginine, since both are known to stimulate pancreatic insulin secretion; our results confirmed that these htNSCs released insulin in response to either stimulation (*Figure 2c*). The insulin-secreting activity of these htNSCs was preserved over at least 10 passages of cell culture which we monitored.

Of interest, compared to cells cultured in the monolayer (*Figure 2b*), insulin secretion increased >30 times by the same number of cells in the spheres (*Figure 2d*), suggesting that the 3D spheroid structure is important for optimizing insulin production and secretion. Also, compared to pancreatic islets, these htNSCs spheres showed a similar pattern of insulin secretion in response to

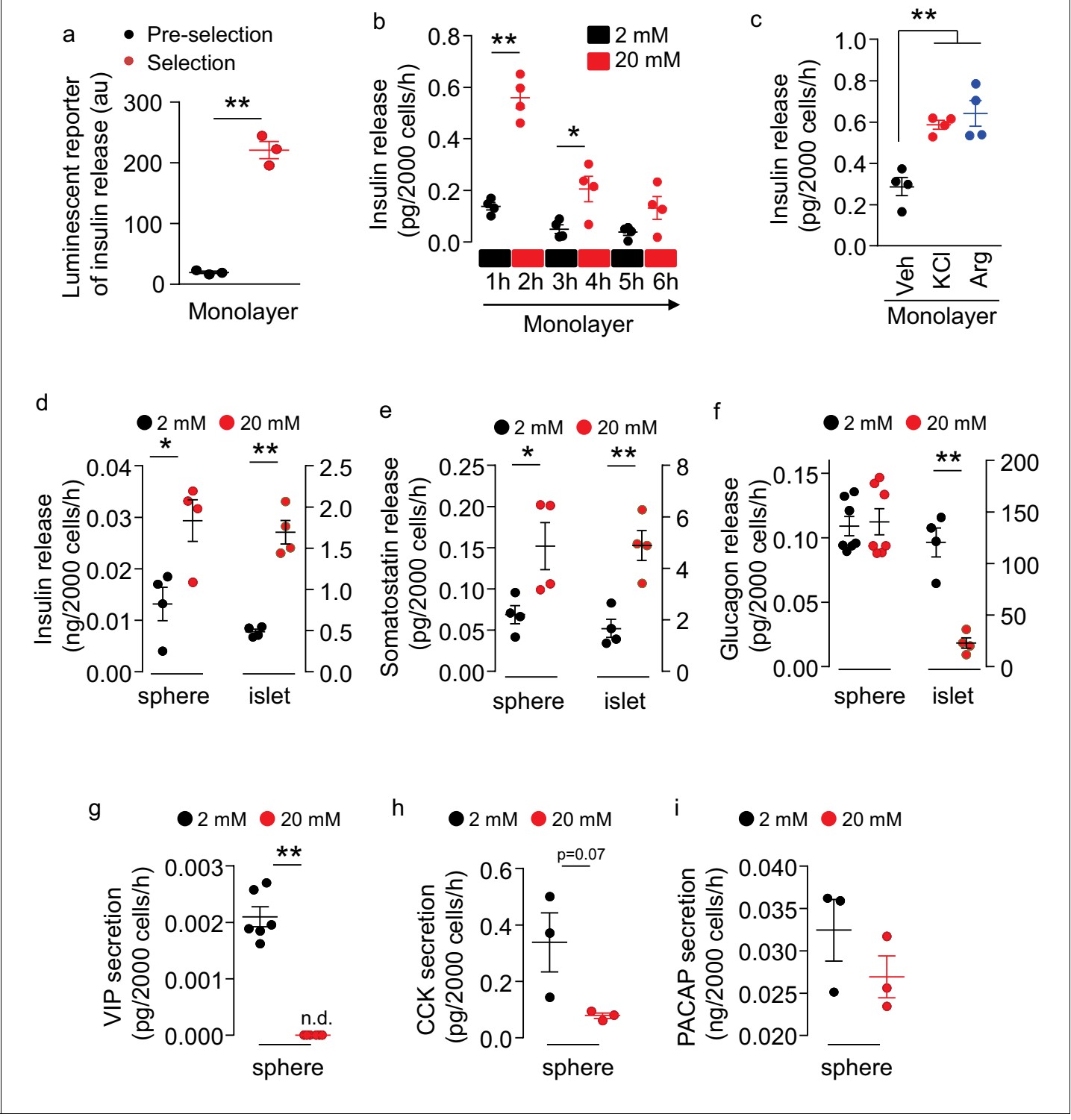

**Figure 2.** Secretion of multiple peptides by insulin promoter-selected htNSC spheres. (**a**) Insulin secretion from insulin promoter-selected htNSCs in monolayer assessed by means of luminescent reporter (au: arbitrary unit). (**b–c**) ELISA measurement of insulin secretion from monolayer culture of insulin promoter-selected htNSCs in serum-free medium under low- vs. high-concentration glucose (**b**) or under stimulation by KCl or Arginine (Arg) vs. vehicle (Veh) (**c**). (**d–f**) Secretion of insulin (**d**), somatostatin (**e**), glucagon (**f**) by mCherry-posiitve htNSC spheres (labeled as spheres) compared to mouse pancreatic islets (labeled as islet). (**g–i**) Secretion of VIP (**g**), CCK (**h**), and PACAP (**i**) by mCherry-posiitve htNSC spheres under low- vs. high-concentration glucose stimulation. *p<0.05, **p<0.01 (Student's t-test for a, g–i, and ANOVA/post-hoc for b–f), n = 3–7 independent biological samples per group (a–i); data are mean ± s.e.m.

glucose, although the capacity was much less comparable to that of pancreatic islets if according to the same number of cells (*Figure 2d*). Besides, some htNSCs in these spheres preserved the features of somatostatin and glucagon secretion, whether they responded to glucose or not (*Figure 2e,f*). Moreover, as exemplified in *Figure 2g–i*, these spheres had the capability of secreting other pancreatic/gastrointestinal peptides such as VIP, CCK and PACAP in glucose-stimulated or glucose-inhibited manners. Thus, we generated a model of htNSC spheres that can secrete multiple pancreatic/gastrointestinal peptides.

## Mixed pancreatic islet-neurosphere features of insulin promoter-selected htNSC spheres

Since these htNSC spheres showed better secretion of insulin compared to its monolayer form, we wondered whether the spherical cellular organization could somewhat structurally resemble pancreatic islets, the majority mass of which are insulin-producing β cells surrounded and interspersed by glucagon-producing α and somatostatin-producing δ cells. We then examined all these pancreatic peptides in these htNSC spheres. Given the heterogeneous sizes of htNSC spheres, we analyzed both large size (*Figure 3a–b*) and small size (*Figure 3—figure supplement 1*), which showed consistent results. For further information, we compared these htNSC spheres to mouse pancreatic islets under the same immunostaining conditions (*Figure 3—figure supplement 2*). As expected, insulin was globally and widely present in these htNSC spheres, which were distributed both in the center and periphery, slightly differing from pancreatic islets in which insulin-positive cells were more in the center than the periphery. Also compared to pancreatic islets, these htNSC spheres contained higher percentages of somatostatin-positive cells but similar percentages of glucagon-positive cells. These cells were probably more established during the sphere development which led to increased promoter activities of these two peptides while insulin promoter activiy became weak and even faded out. Also, while glucagon- and somatostatin-positive cells were distributed typically in the periphery of pancreatic islets (*Figure 3—figure supplement 2a–b*), it was the case for somatostatin but not for glucagon in these htNSC spheres (*Figure 3a–b*, *Figure 3—figure supplement 1a–b*). Besides, these htNSC spheres expressed other pancreatic and gastrointestinal hormones such as PACAP and CCK, which seemed stronger compared to these peptides in pancreatic islets. In addition, we examined pancreatic islet biomarkers including proprotein convertase 1 (PC1/3), glucose transporter 2 (GLUT2), ATP-sensitive K$^+$ channel KIR6.2, and NKX2.2. The results showed that all these biomarkers were widely expressed in these htNSC spheres (*Figure 3a–b*, *Figure 3—figure supplement 1a–b*), and each pattern was pretty similar to that in pancreatic islets (*Figure 3—figure supplement 2*). On the other hand, unlike the pancreatic islets which barely expressed NSC biomarkers Sox2 and nestin, both biomarkers were strongly and universally expressed in these htNSC spheres (*Figure 3a–b*, *Figure 3—figure supplements 1–2*), suggesting that this hypothalamic spherical model still kept the key features and molecular signatures of htNSCs.

## Requirement of somatostatin- and glucagon-positive cells for htNSC spherical structure

Recent studies revealed that electrical and structural coupling of different cell types in pancreatic islets could be important for the metabolic function of islets (*Hoang et al., 2014*; *Briant et al., 2018*). We hypothesized that that strategy of developing β cell therapy against diabetes might need to consider the importance of glucagon-producing and somatostatin-producing cells in 3D structure. In order to provide an insight into this idea, we determined the inter-relationships of insulin-, somatostatin- and glucagon-expressing cells in maintaining the 3D structure of insulin promoter-selected htNSC spheres. To do so, we employed promoter-driven diphtheria toxin (DT) receptor (DTR) which led to selective depletion of insulin-, somatostatin- or glucagon-positive cells in these htNSCs spheres after DT administration in the culture medium. Without surprise, htNSC spheres were completely damaged when either cell type was depleted. We then collected the dissociated cells from this DT treatment by which either of individual cell types was mostly eliminated, and examined if the remaining cells could eventually lead to spheres. Of interest, when insulin-positive cells were mostly missing, the remaining cells still effectively made spheres; in contrast, when either somatostatin- or glucagon-positive cells were mostly missing, despite that a lot of insulin-producing cells were present, these cells failed to form any spheres (*Figure 3c*). Thus, the co-presence of

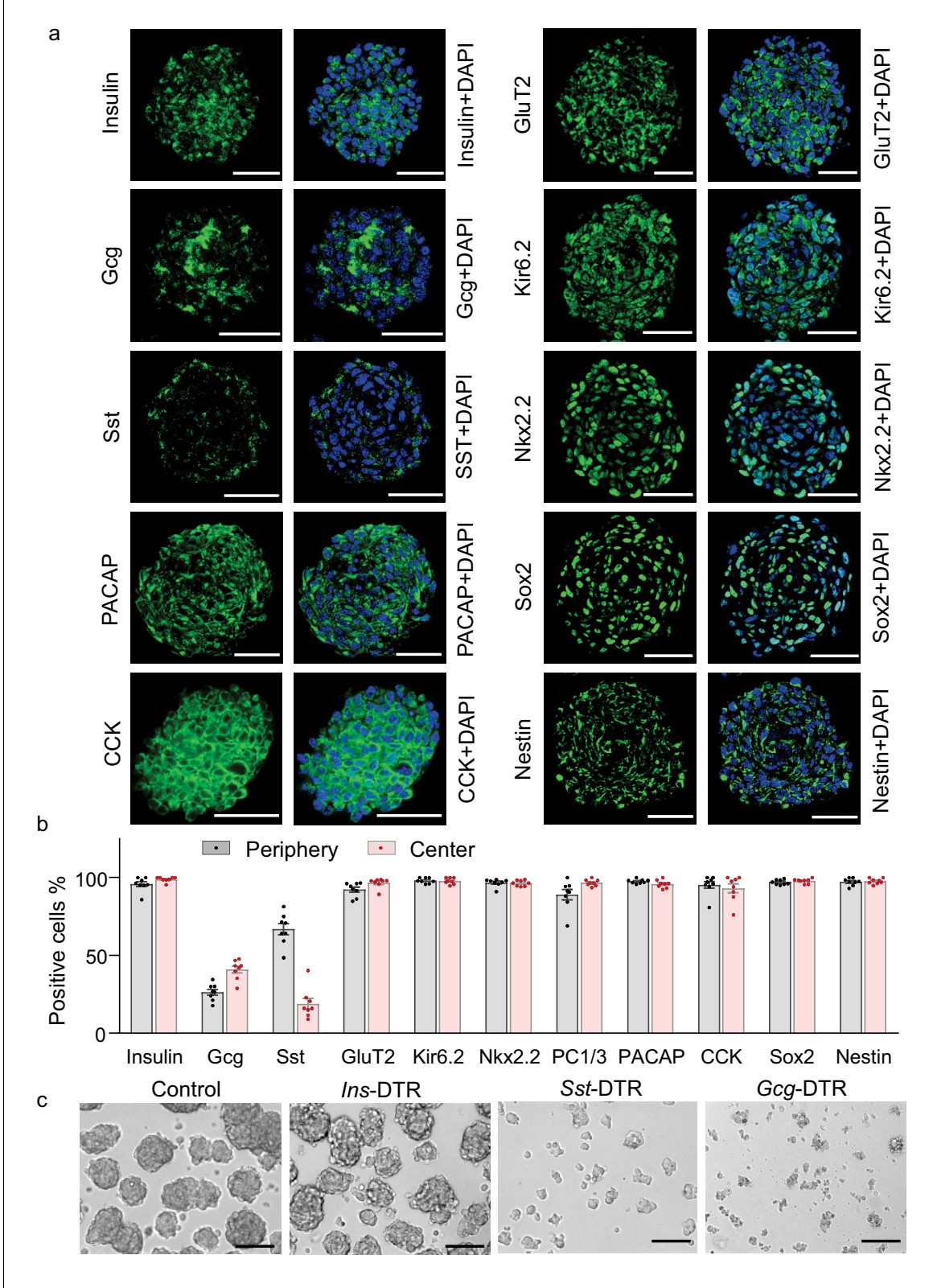

**Figure 3.** Multifaceted endocrine features of insulin promoter-selected htNSC spheres. (a–b) Cultured insulin promoter-selected htNSC spheres were immunostained for molecular markers of pancreatic islets and hypothalamic stem cells (a), and percentage of positive cells located within the center versus the periphery of spheres were calculated (b). All data in (a–b) were based on large-sized htNSC spheres (please see *Figure 3—figure supplement 1* for small-sized htNSC spheres). Sst: somatostatin; Gcg: glucagon. Scale bar, 50 μm. (c) Insulin-, Sst- and Gcg-positive cells in insulin

*Figure 3 continued on next page*

*Figure 3 continued*

promoter-selected htNSC spheres were respectively ablated through lentiviral induction of diphtheria toxin receptor (DTR) driven by insulin (*Ins-*), somatostatin (*Sst-*) or glucagon (*Gcg-*) promoter and followed by 24 hr diphtheria toxin treatment, and the same number of surviving cells in each group were collected for forming spheres over 72 hr. Scale bar: 100 μm. Data in (b): n = 8 independent biological samples per group; data are mean ± s.e.m.

The online version of this article includes the following figure supplement(s) for figure 3:

**Figure supplement 1.** Multifaceted features of small-sized htNSC<sup>PGHM</sup> spheres.
**Figure supplement 2.** Immunostaining of pancreatic islets.

somatostatin- and glucagon-positive cells is crucial for making spheres and keeping insulin-positive cells.

## Abundance of miRNA exosomes in insulin promoter-selected htNSC spheres

Given that recently we reported that htNSCs have a unique ability to abundantly secret miRNAs-containing exosomes, which suggested a new type of hypothalamic endocrine function (*Zhang et al., 2017*), we then evaluated if this key endocrine feature was retained in this model of insulin promoter-selected htNSC spheres. Using exosomal biomarkers CD81 and TSG101, we performed immunostaining for these htNSC spheres versus pancreatic islets. As shown in *Figure 4a–b*, we confirmed that both CD81 and TSG101 were strongly expressed in >90% of cells in these htNSC spheres, but these biomarkers were barely detected in pancreatic islets. Furthermore, we isolated exosomes from these htNSC spheres versus pancreatic islet cells and subjected them to the assay by small RNA bio-analyzer. As shown in *Figure 4c–d*, these htNSC spheres secreted a considerably high amount of exosomal miRNAs while pancreatic islet cells barely did so. Thus, this model of htNSCs spheres retained the unique function of htNSCs in releasing miRNAs-containing exosomes. Hence, given that these htNSC spheres can produce pancreatic and gastrointestinal as well as hypothalamic peptides in addition to miRNAs-containing exosomes, henceforth these spherical cells were named as htNSC<sup>PGHM</sup> and these hypothalamic stem cells-derived spheres were also referred to be hypothalamic islets.

## Peripheral implantation of htNSC<sup>PGHM</sup> spheres in STZ mouse model

Having appreciated the secretion of pancreatic hormones and the structural similarity compared to pancreatic islets, we intuitively asked if htNSC<sup>PGHM</sup> spheres could have a therapeutic effect against insulin-deficient diabetes. Given our focus on endocrine secretion as well as our interest in practical application, we designed to explore several peripheral locations which have often used for cell implantation in research, including subcapsular spaces in kidneys, subcutaneous and intrahepatic spaces. To help the in vivo survival of htNSCs as we established previously (*Li et al., 2012*), these htNSC<sup>PGHM</sup> cells were engineered via lentiviral induction to stably contain a potent anti-inflammatory gene, dominant-negative IκBα (GFP-conjugated, controlled by CMV promoter). However, despite these efforts, none of these peripheral locations turned out to be successful for the implantation of htNSCs due to the poor survival when ectopically implanted. As we continued to try other peripheral locations, we found that the great omentum was a suitable choice of implantation, with >50% implanted cell survival over a month. The great omentum is a peritoneum membrane of the abdominal cavity, which is mainly composed of fat and connective tissue and has abundant blood supply. In mice, the great omentum is mostly curled under the stomach's greater curvature and can be spread up to provide a good physical support for implants. In our procedure, as indicated in *Figure 5a*, htNSC<sup>PGHM</sup> spheres were embedded into 3D collagen matrices while solidified collagen matrices were compressed into collagen sheets with tension and rigidity similar to in vivo tissues, and cell-embedded collagen sheets were placed and secured into the great omentum flap of mice.

To profile if implantation of htNSC<sup>PGHM</sup> spheres could provide an anti-diabetic effect, we employed streptozotocin (STZ) mouse model. Administration of STZ, an alkylating chemical that is well known for toxicity on pancreatic β cells, has been well established to cause severe hyperglycemia and mortality in mice. We adopted a protocol of single high-dose STZ injection to generate a severe diabetes and sickness model. Hyperglycemia (>350 mg/dL) was induced in ~80% of mice

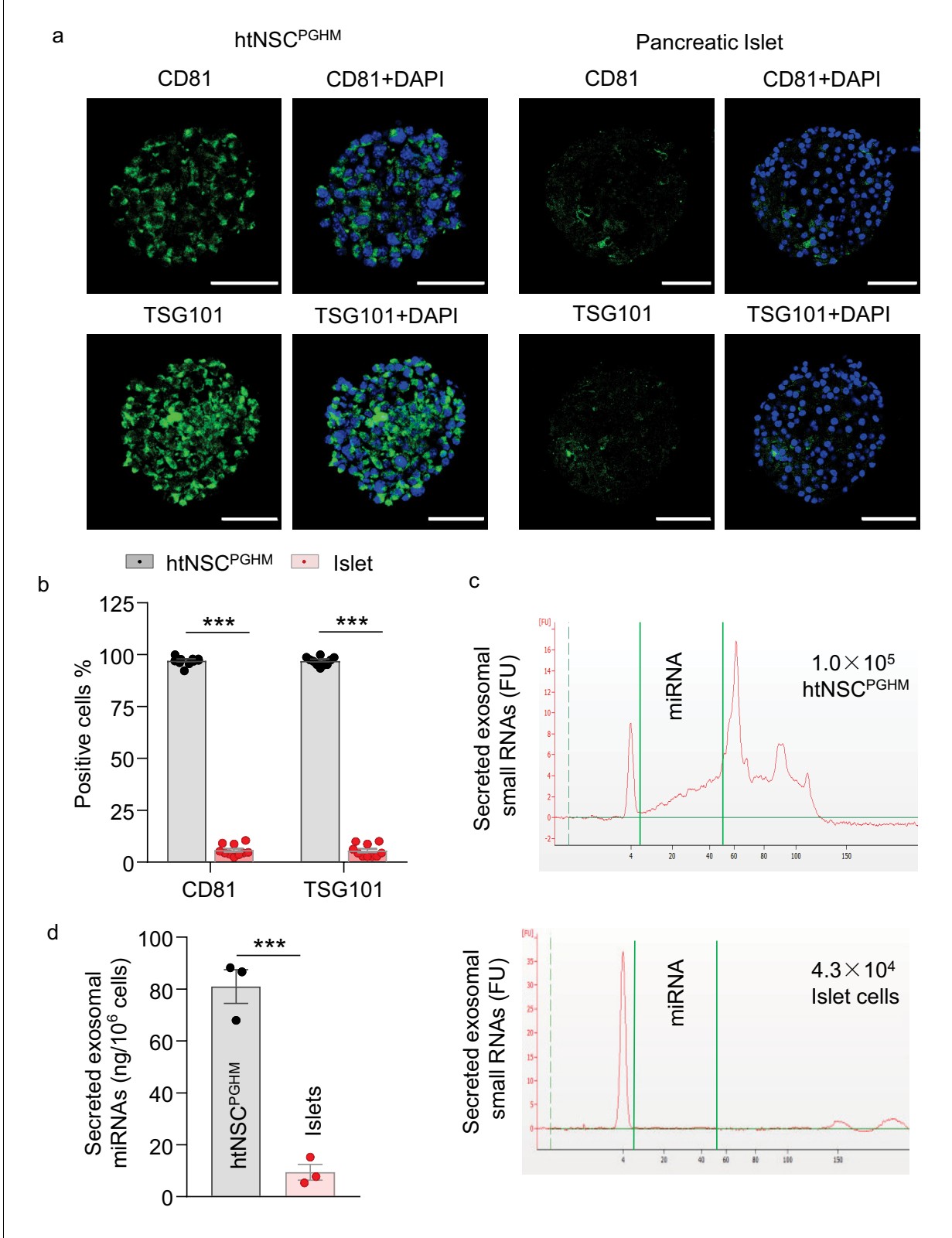

**Figure 4.** Exosomes in insulin promoter-selected htNSC spheres vs. pancreatic islets. (a–b) Cultured insulin promoter-selected htNSC spheres vs. pancreatic islets obtained from mice (3-month-old C57BL/6J males) were immunostained for molecular markers of exosomal biomarkers (a), and percentage of cells positive for each biomarker over the entire spherical or islet cells were calculated (b). Scale bar, 50 μm. (c–d) Exosomal small RNA and miRNA secreted by insulin promoter-selected htNSC spheres vs. pancreatic islets. (c): Representative tracing; (d) Quantification of exosomal

*Figure 4 continued on next page*

*Figure 4 continued*

miRNAs secreted by insulin promoter-selected htNSC spheres vs. islets according to the same number of cells. ***p<0.001 ( ANOVA/post-hoc for b, and student's t-test for d), n = 10 (**b**) and n = 3 (**d**) independent biological samples per group, data are mean ± s.e.m.

within 5–7 days after STZ administration, and roughly 50% of these mice died within 30–40 days (which slightly varied among different experiments when involving different procedural stresses). As represented in *Figure 5a*, these STZ-injected diabetic mice were implanted with htNSC[PGHM] spheres in the great omentum vs. sham implantation (a group of mice without STZ injection was also included as a normal reference). Prior assessments also included subgroups of mice from this procedure to provide cell implants which were sectioned for the analysis of cell survival. As shown, about 50% of implanted cells survived over 1 month (*Figure 5b*), and these surviving cells were also positive for insulin promoter-driven mCherry (*Figure 5c*). Thus, we generated a model of hypothalamic islets in the periphery of the body when pancreatic islets were damaged, which allowed us to study if this model has any therapeutic significance.

## Moderate antidiabetic effect of htNSC[PGHM] spheres in STZ model

We examined whether the ectopic implantation of htNSC[PGHM] islets could treat diabetes in the STZ mouse model. As shown in *Figure 5d*, while overnight fasting blood glucose remained 350–500 mg/dL in majority of sham-operated STZ mice, blood glucose dropped below 300 mg/dL in a majority of mice with htNSC[PGHM] implantation shortly after implantation, and many of them were lower than 250 mg/dL. These therapeutic effects remained for weeks or throughout the follow-up. Although a portion of animals (~25% of experimental mice) did not show an appreciable hypoglycemic effect, it was revealed at the end of the experiment that cells in the implants survived poorly in these mice. However, these hypoglycemic effects were considered moderate since these treated mice were still diabetic. Because insulin is an important factor for lowering blood glucose, and our generation of htNSC[PGHM] was indeed based on insulin promoter for cell selection, we measured fasting plasma insulin levels of the implanted mice. As shown in *Figure 5e*, average plasma insulin levels in cell-implanted STZ mice were in general higher than the levels in sham controls, although they were still much below the reference of normal mice. Because the primary target of STZ is pancreatic islets, a relevant question was whether htNSC[PGHM] spheres could induce regeneration of pancreatic islets. Through rigorous examination, we confirmed that pancreatic islets were similarly damaged between cell implantation and sham controls (*Figure 5f,g*), indicating that the effect of this treatment was not due to pancreatic islet protection in this STZ model. Despite that htNSC[PGHM] were much less potent than pancreatic islets in releasing insulin, we had the advantage of implanting a large number of spheres (>100 times more than the number of pancreatic islets of a mouse), which should be significant for elevating plasma insulin and thus helped diabetic control in these STZ mice.

## Remarkable pro-survival effect of htNSC[PGHM] islets in STZ model

As expected, STZ-administrated mice with the sham operation displayed weight loss and severe sickness such as hypothermia, hunched appearance and inertia leading to death in majority of these mice. In contrast, STZ mice with successful implantation of htNSC[PGHM] islets showed great resistance to all these sickness phenotypes. Because survival rate is an important indicator for the severity of fatal disease, we followed up a group of htNSC[PGHM]-treated mice for their survival over an 80 day duration. Our results showed that mouse survival rate remarkably increased by this treatment (*Figure 6a*). While the majority of death occurred within 40 days after STZ injection in the sham group, death was greatly avoided when htNSC[PGHM] islets were implanted. Improvement of general health by this treatment was also reflected by improved body weight (*Figure 6b*). For these mice over survival follow-up, we periodically measured blood glucose and insulin levels and examined if these metabolic profiles could be correlated with their survival. As shown in *Figure 6c*, there was a lack of correlation between hyperglycemia and survival in sham STZ model, suggesting that death of STZ mice was not simply a result of hyperglycemia when blood glucose levels were very high. While the sphere-treated STZ mice had a moderate reduction in hyperglycemia (they were still diabetic) and a much greater improvement of survival, the correlation analysis revealed only a tendency of relationship between blood glucose and survival (*Figure 6d*), indicating that the limited benefit of

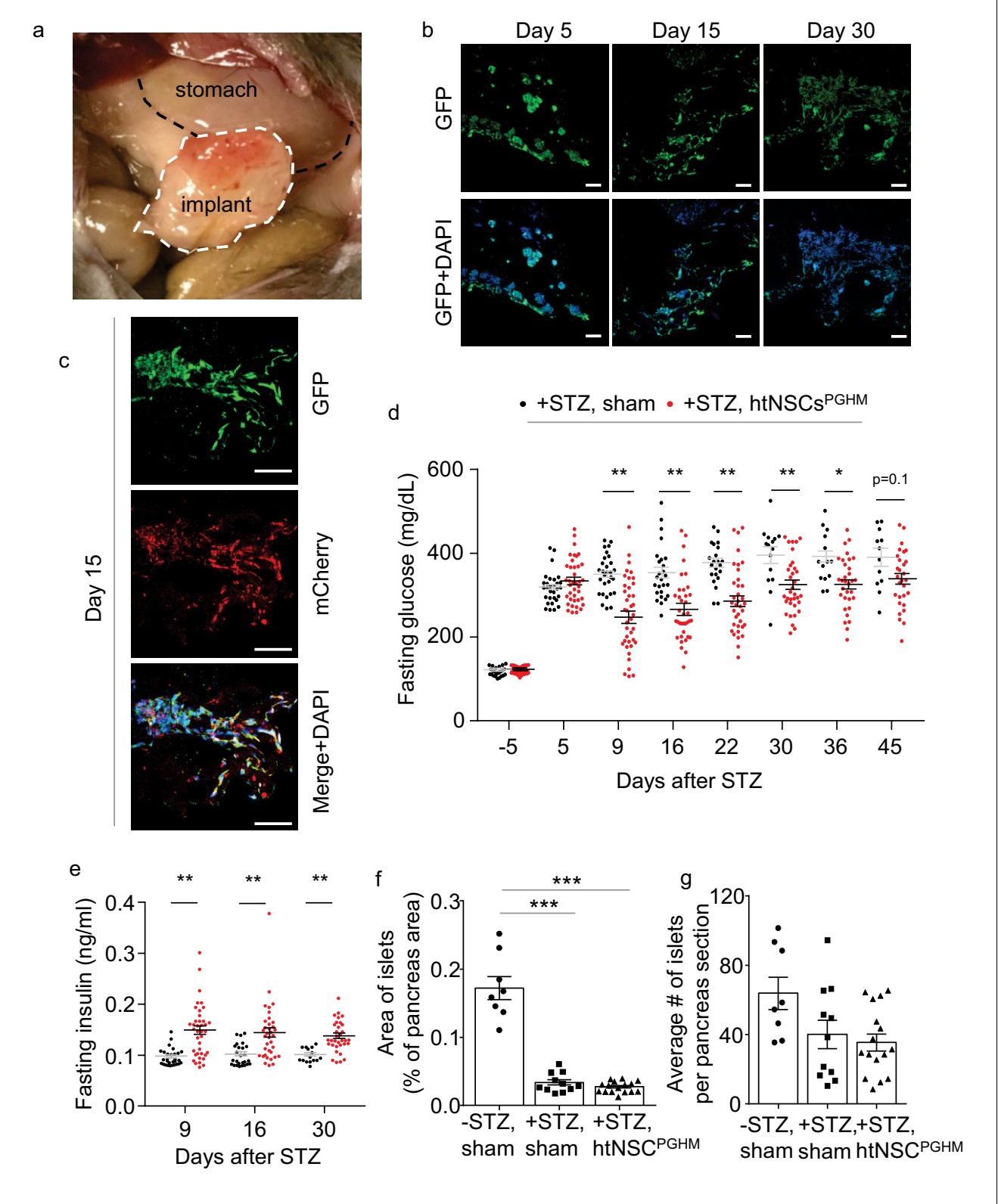

**Figure 5.** Moderate antidiabetic effect of htNSCs^PGHM spheres in STZ model. (a–c) Implants of htNSC^PGHM in the great omentum of STZ mice were extracted and sectioned for examining cell survival. (a): representative implantation; (b–c): representative low-magnification sectional images of GFP expression at the indicated days post implantation (b) and representative high-magnification sectional images of GFP (controlled by CMV promoter) and mCherry (controlled by insulin promoter) co-expression (c). Scale bar, 100 μm. (d–e) Standard C57BL/6J mice (3-month-old males) were injected by

*Figure 5 continued on next page*

*Figure 5 continued*

STZ to develop severe diabetes (+STZ) and then received htNSC^PGHM implantation (+STZ, htNSC^PGHM) vs. sham implantation (+STZ, sham) at days 5–8, and survived mice were monitored for fasting blood glucose (d) and insulin (e) at the indicated time points. Survival mice in sham vs. htNSC^PGHM group were n = 30, 30, 29, 26, 22, 15, 14 and 12 vs. n = 39, 39, 39, 37, 36, 34, 32 and 29 at day −5, 5, 9, 16, 22, 30, 36 and 45, respectively. (f–g) Separate groups of sham-implanted STZ mice (+STZ, sham; n = 11 mice) vs. htNSC^PGHM-implanted STZ mice (+STZ, htNSC^PGHM; n = 16 mice) with the same procedure and conditions as described in d-e were compared to sham-implanted normal control mice (-STZ, sham, n = 8 mice) for the histology of the pancreas (day 40 post implantation). *p<0.05, **p<0.01, ***p<0.001, (ANOVA/post-hoc); data are mean ± s.e.m.

blood glucose drop might not be most important for the improved survival of these mice. In terms of blood inulin, there was also a lack of clear correlation between its level and the survival in sham STZ mice (*Figure 6e*). However, in sphere-treated STZ mice, the correlation between insulin and survival was marginally significant (*Figure 6f*), suggesting that elevation in blood insulin could have a partial contribution to the improved survival of these mice. We also subjected a group of implanted mice to a battery of behavioral tests, and as shown in *Figure 6g–i*, spheres-implanted STZ mice showed much less declines in muscle strength, treadmill running capability and voluntary locomotion, compared to STZ sham mice.

## Metabolic versus pro-survival effects by htNSC^PGHM islets involving different secretion

We further investigated if the pro-survival and health-improving effects of htNSC^PGHM in STZ diabetes model could be secondary to the metabolic improvement of this treatment. To do so, we generated a clone of htNSC^PGHM in which insulin expression was genetically disrupted. In the protocol, htNSC^PGHM (as described in *Figures 5–6*) were infected with lentiviral shRNAs against *Ins2* gene leading to an htNSC cell line for spheres named htNSC^PGHM-shIns. Somatostatin or glucagon was not targeted, since we found that they are important for forming spheres (*Figure 3c*). For comparison, we knocked down two additional pancreatic/gastrointestinal peptides by infecting htNSC^PGHM with shRNA lentiviruses against three genes including *Ins2*, *Pacap* and *Cck,* and this htNSC cell line leading to spheres referred as to htNSC^PGHM-shIPC. To target exosomes, htNSC^PGHM were infected with Rab27a shRNA lentiviruses, a method which inhibits exosomal release of htNSCs as we evaluated in our recent work (*Zhang et al., 2017*), and this htNSC cell line leading to its spheres was referred as to htNSC^PGHM-shRab. For the control, infection with scramble shRNA lentiviruses led to htNSC^PGHM-shCtrl cell line and its spherical model.

These different groups of shRNA-engineered htNSC^PGHM islets were implanted into STZ mice, which were monitored for health, survival and metabolic profiles. Consistently, htNSC^PGHM-shCtrl islets provided moderate effects in lowering blood glucose and elevating plasma insulin (*Figure 7a, b*), although these effects seemed slightly better compared to previous experiments which was likely due to our procedural improvements. In contrast, due to inhibition of insulin in these spheres, the glucose-lowering effect of this therapy was completely abrogated as observed in both htNSC^PGHM-shIns and htNSC^PGHM-shIPC groups (*Figure 7a,b*). These results suggest that an elevation of plasma insulin due to implanted htNSC^PGHM islets accounted for the moderate hypoglycemic effect of the treatment. To our surprise, these spheres still provided a dramatic although partially reduced effect in improving survival and general health of these STZ mice (*Figure 7c*) despite that diabetes in these mice was same severe as was STZ control (*Figure 7a*). On the other hand, exosomal inhibition in these spheres as represented in htNSC^PGHM-shRab group did not significantly affect the glucose-lowering or insulin-elevating benefits of the therapy (*Figure 7a,b*), but it completely abrogated htNSC^PGHM spheres from improving the survival of STZ mice (*Figure 7c*). In addition to survival follow-up, we further behaviorally assessed the general health of these STZ mice under the treatment of these different spheres. As shown in *Figure 7d–g*, while the treatment of control spheres did greatly improve all these behavioral functions compared to sham-operated STZ group, these improvements only marginally compromised when insulin in these spheres was inhibited but were completely abrogated when exosomes in these spheres were inhibited. Therefore, htNSC^PGHM spheres have differential roles in improving metabolic physiology versus general health, and multifaceted secretion of these spheres are mechanistically important for the different aspects of these therapeutic benefits.

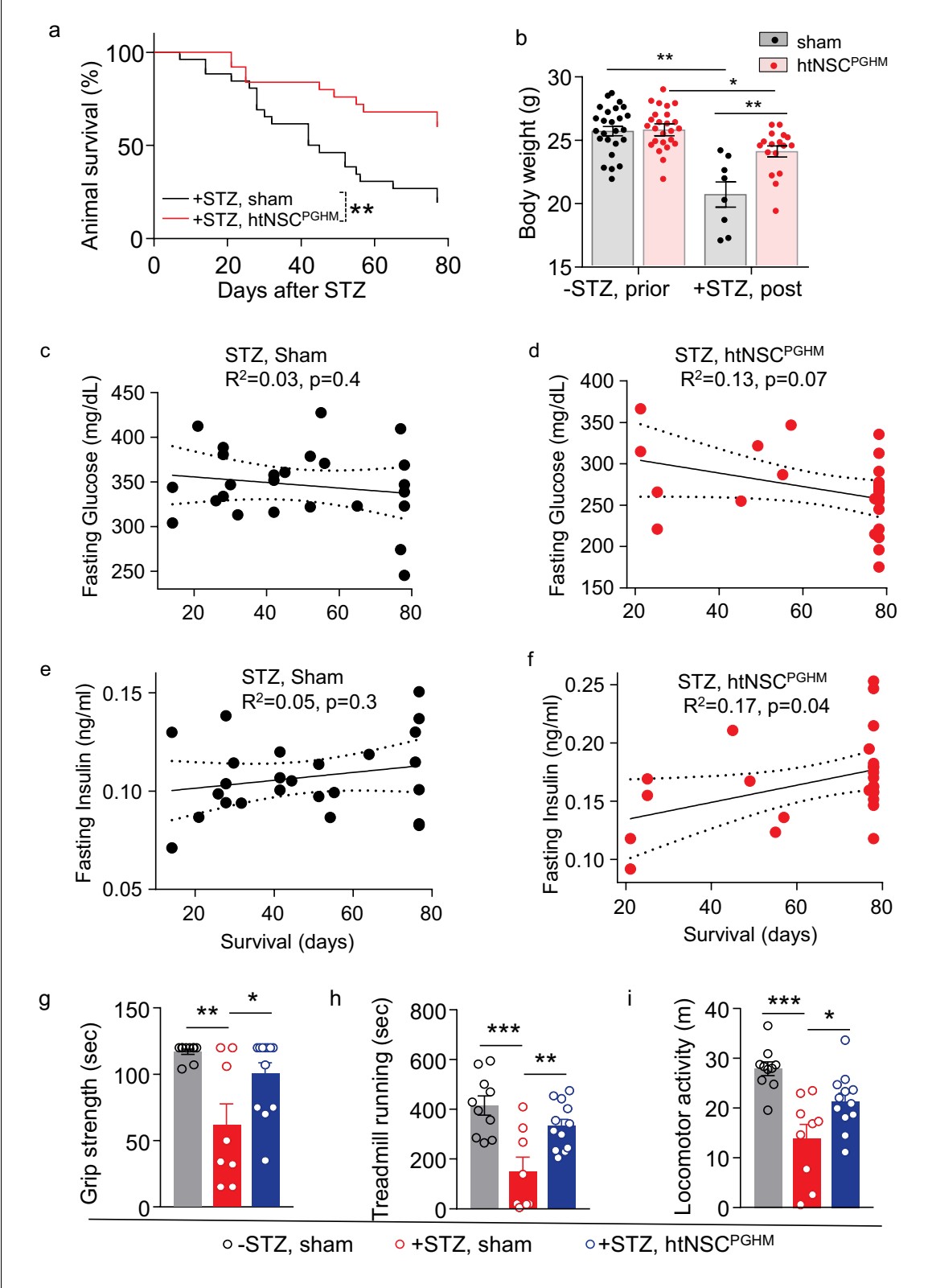

**Figure 6.** Pro-survival effect of htNSCs[PGHM] spheres in STZ model. (a–f) STZ mice received htNSC[PGHM] implantation (+STZ, htNSC[PGHM], n = 25 mice at day 0) vs. sham implantation (+STZ, sham, n = 26 mice at day 0) in the great omentum. (a): survival curve of mice over 80 day follow-up; (b): body weight of mice before (-STZ) and after (+STZ) induction of STZ diabetes prior to vs. 8 weeks post htNSC[PGHM] vs. sham implantation (n = 25 vs. 26 mice for '-STZ, prior', and n = 17 vs. 8 mice for '+STZ, post'); (c–f) The correlation analysis between days of survival and blood glucose and insulin levels

*Figure 6 continued on next page*

Figure 6 continued

averaged within two weeks after implantation. (g–i) Separate groups of sham-implanted STZ mice (+STZ, sham; n = 8–9 mice) vs. htNSC^PGHM-implanted STZ mice (+STZ, htNSC^PGHM; n = 12 mice) were compared to sham-implanted normal control mice (-STZ, sham, n = 10 mice) for non-invasive behavioral performance at 3–4 weeks post implantation. *p<0.05, **p<0.01, ***p<0.001, Mantel-Cox survival test (a), linear regression (c–f), and ANOVA/post-hoc (b, g–i); data are mean ± s.e.m.

## Discussion

The hypothalamus contains neurons which express various peptidyl hormones including some pancreatic/gustatory hormones, and in fact, secretion of individual peptides has been employed to classify hypothalamic neuronal subtypes. Of course, the pancreatic islets have critical functions of secreting insulin, glucagon, somatostatin and several other peptides, and secretion of these hormones has often been investigated according to specific cell types in the islets. Similarities exist between the hypothalamus and pancreatic islets, for instance, the hypothalamus is the major site of insulin expression other than islets of Langerhans in the pancreas, suggesting that these organs have a particular relationship. In this study, we reported that htNSCs contain subpopulations that produce multiple secretory factors including a variety of pancreatic/gastrointestinal and hypothalamic peptides as well as miRNAs-containing exosomes. Although secretion of a particular peptide by htNSCs is much less strong compared to the committed mature cells such as endocrine neurons or pancreatic endocrine cells, multifaceted secretion of htNSCs represents a unique feature with potential wide-ranged functions which could be a biological basis for various applications. Using insulin promoter activity as a lead of selection, we established a subpopulation(s) of htNSCs and derived these cells into the spherical structure which contain combined features of hypothalamic neurospheres and pancreatic islets. We proved that the 3D spherical structure is critical for releasing multiple hormones including insulin, and in particular, the co-existence of somatostatin- and glucagon-positive cell subtypes is necessary to form and maintain these spheres. This finding could be suggestive to the research interest in attempt to generate functional β cells from stem cells, as spherical structure and heterogeneity of cell components seem to be important for this success.

Because our hypothalamic spheres impressively mimic the feature of pancreatic islets, we examined if they could provide a therapeutic effect against insulin-deficient diabetes. We successfully developed the approach of peripherally delivering these hypothalamic spheres in mice and applied them to an experimental model of STZ-induced fatal diabetes. Thus, when pancreatic islets were damaged in mice, we came up with a substitution through using hypothalamic islets. Surprisingly the treatment from this peripheral implantation led to dramatic improvements in general health and survival although the antidiabetic effect was moderate. We further revealed that this pro-survival benefit was dissociable from blood glucose control to a significant extent. Based on this observation, we suggest that diabetic management should require consideration over health-enhancing endocrine factors in addition to blood glucose control. Also, although the antidiabetic effect was relatively moderate in the experimental model of STZ-induced extreme diabetes, it is likely that this therapy could be more appreciable for etiological forms of diabetes which are generally much less severe. We further speculate that the antidiabetic effect of these cells might provide an explanation for the recently-appreciated central FGF treatment in leading to sustained diabetic remission (*Scarlett et al., 2016*), given that htNSCs contain tanycytes which proliferate in response to FGF stimulation (*Robins et al., 2013*), and indeed FGF was notably important for maintaining and growing the hypothalamic spheres in our study.

To summarize, a subpopulation(s) of multi-secretory htNSCs can be developed to generate 3D spheres with combined features of neurospheres and pancreatic islets, and when these hypothalamic islets are implanted peripherally, the treatment provides a wide range of benefits including the protection against mortality and diabetic symptom. In this context, we suggest that this therapeutic trial can be extended to other fatal health problems which might not necessarily be related to diabetes. Certainly there are many outstanding questions, for example, how this technology could be translated into human application, whether a practical source of cells such as iPS could be used for generating these spheres, and even if multifaceted secretion of this spherical model could be mimicked by an artificial system, all of which warrant future investigation.

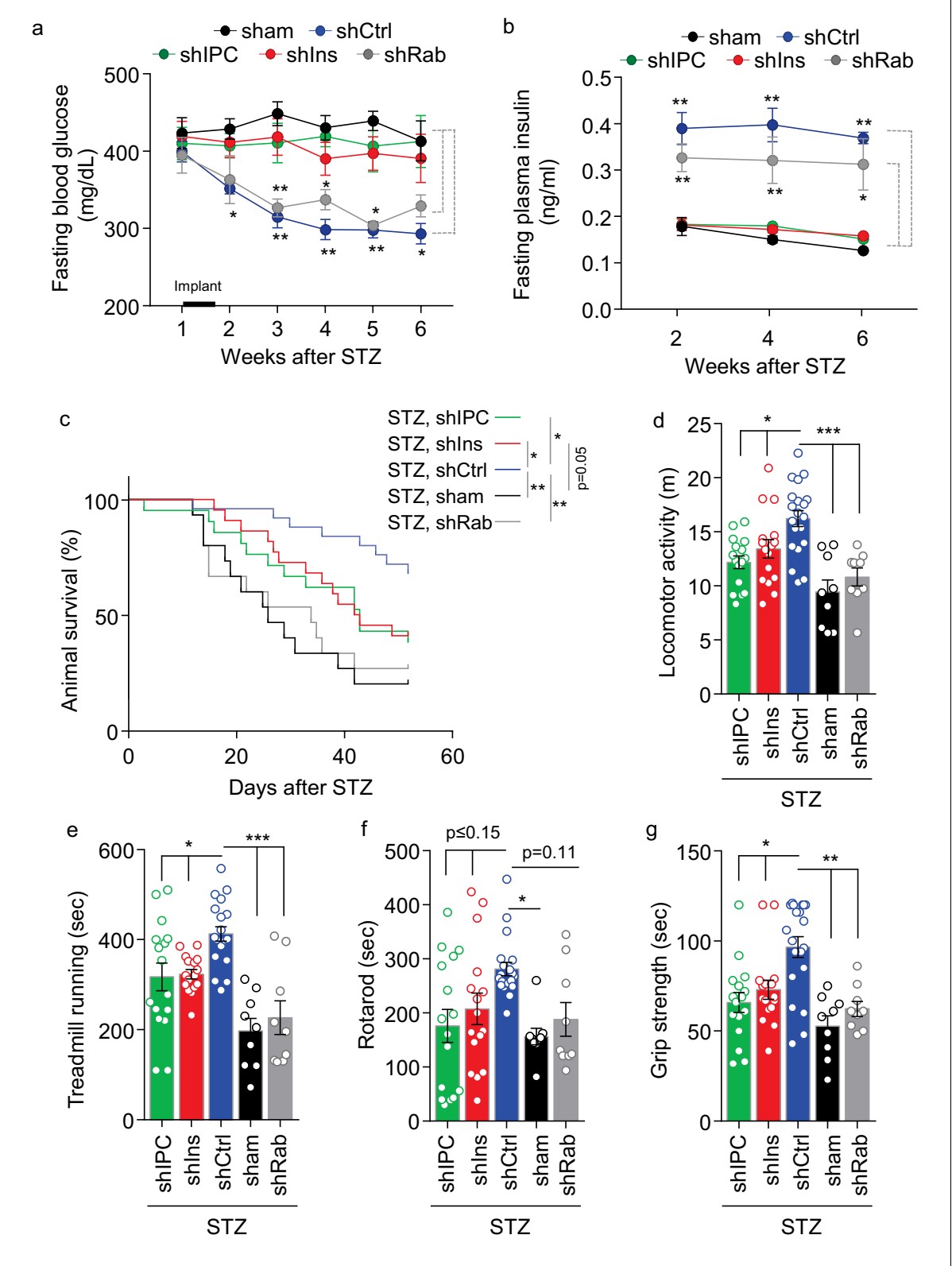

**Figure 7.** Role of multi-secretion of htNSC^PGHM spheres for its anti-disease effects. STZ mice received sham implantation (+STZ, sham) vs. implantation with htNSC^PGHM expressing shRNAs against insulin (shIns), triple peptides including insulin, PACAP and CCK (shIPC) or Rab27a-positive exosomes (shRab) vs. scramble shRNA control (shCtrl). The mice at the starting point of this experiment were n = 15, 22, 21, 15 and 25 per group, respectively. These mice were periodically measured for fasting blood glucose (**a**) and insulin (**b**) while being followed up for survival (**c**), and subgroups of mice were

*Figure 7 continued on next page*

*Figure 7 continued*

subjected to non-invasive behavioral assays at 4 weeks post implantation (d–g). Changes of studied sample sizes over time in (b) and (c) were indicated by the survival rates in a. The studied sample sizes in (d–g) were n = 16–20 mice per group in shIPC, shIns, and shCtrl, and n = 9 mice per group in shRab and sham. *p<0.05, **p<0.01, ***p<0.001, ANOVA/post-hoc (a–b, d–g), and Log-rank (Mantel-Cox) survival test (c); compared between time-matched groups indicated by dash lines (a, b), or between indicated curves (c) or columns (d–g); data are mean ± s.e.m.

## Materials and methods

### DNA constructs and recombinant viruses

Plenti6.3TOV5DEST (Invitrogen) backbone was used to construct the dual Cre/Flp recombinase lenti-viral reporter plasmid. A multi-cloning site was inserted into the SpeI/MluI-digested backbone, generating plenti6.3TOMCS. lox-STOP-lox (LSL) sequence was from Ai65(RCFL-tdT) targeting vector (Addgene# 61577). EF1a-mCherry sequence was from pSicoR-Ef1a-mCh-Puro-*GFPi* (Addgene# 31848). frt-neoSTOP-frt-*EGFP*-pA sequence was from pCAFNF-*GFP* (Addgene # 13772). A lentiviral vector harboring RIP (rat insulin promoter)-driven mCherry expression cassette was also constructed on plenti6.3TOMCS backbone, and the *RIP* sequence was from pLOX-IWgfp (Addgene # 12244). The generated vector was named RIPmCherry. The cDNA sequences of Cre (from Addgene plasmid#13775), codon-optimized *Flp* (FlpO, from Addgene plasmid#13793), DTR (*Zhang et al., 2017*), and PCR-amplified promoter sequences (for rat insulin 2, mouse glucagon and somatostatin) were inserted into CMV promoter-deleted plenti6.3TOMCS, generating *Ins*-Cre, *Ins*-FlpO, *Ins*-DTR, *Gcg*-Cre, *Gcg*-FlpO, *Gcg*-DTR, *Sst*-Cre, *Sst*-FlpO and *Sst*-DTR lentiviral vectors. Plasmids of pAd_Ngn3-I-*nGFP* (Addgene #19410), pAd_PdxI-I-*nGFP* (Addgene #19411) and pAd_MafA-I-*nGFP* (Addgene #19412) were purchased from Addgene. All shRNA lentiviral plasmids for individual genes were purchased from Mission lentiviral shRNA Library (Sigma). All plasmids were verified through restriction digestion and DNA sequencing. Recombinant adenoviruses were generated in HEK293 cells, purified and concentrated with Fast-Trap Adenovirus purification and concentration kit (Millipore), and titrated with Adeno-X Rapid Titer Kit (Clontech) according to the manufacturer's instructions. Generation, concentration, and titration of lentiviruses have been described previously (*Zhang et al., 2008*). Briefly, HEK293T cells were co-transfected with the lentiviral vector and the packaging plasmids. The cell debris-removed, viral particle-containing culture medium was concentrated by ultracentrifugation at 82,000 g for 2 hr. The recombinant lentiviruses were reconstituted in a small volume of PBS to achieve a 1000X concentration, and titrated using p24 ELISA kit (ZeptoMetrix) according to the manufacturer's instruction.

### Cell culture and models

Primary culture of htNSCs was performed as similarly described previously (*Esposito et al., 2012*). In brief, the hypothalamus was dissected from newborn C57BL/6J mice, cut into pieces of approximately 1 mm³, and followed by digestion using TrypLE Express enzyme (Life Technologies) for 10 min at 37°C. After washes, cells were suspended in the NSC medium composed of neurobasal-A (Life Technologies), 0.25% GlutaMAX supplement (Life Technologies), 2% B27 without vitamin A (Life Technologies), 10 ng/ml EGF (Sigma-Aldrich),10 ng/ml bFGF (Life Technologies) and 1% penicil-lin–streptomycin and seeded in Ultra-Low Attachment 6-well plates (Corning). One week later, neu-rospheres were collected by centrifugation and trypsinized with TrypLE Express enzyme into single cells, passaged and maintained in neurosphere culture until experimental use. HEK293T cells were purchased from ATCC (CRL-3216) and were cultured in Dulbecco's modified Eagle medium (DMEM) containing 10% fetal bovine serum and 1% penicillin-streptomycin. All cell lines used in this study were free of microbial (including mycoplasma) contamination and their morphology and growth characteristics were compared to published information to ensure their authenticity. IκBα-htNSCs cells were generated as previously described (*Li et al., 2014*). Cultured htNSCs cells were infected with RIPmCherry lentiviruses and Pdx-1, Ngn3, and MafA adenoviruses for two days, and then disso-ciated into filtration-sterilized sorting buffer containing 1x Phosphate Buffered Saline (Ca/Mg++ free), 2 mM EDTA, 25 mM HEPES pH 7.4, 1% Fetal Bovine Serum (Heat-Inactivated), 2% B27, and 1% penicillin-streptomycin. The mCherry+ cells at single-cell purity were isolated with MoFlo XDP cell sorter (Beckman Coulter) by the flow cytometry core facility of Albert Einstein College of Medicine. Cell ablation: htNSCs were infected with *Ins*-DTR, *Gcg*-DTR or *Sst*-DTR lentivirus for 48 hr

and were subjected to 5 ng/mL of diphtheria toxin (DT) treatment for 24 hr. The post-treatment surviving cells were counted and re-seeded at the density of $1 \times 10^5$ cells/mL to form spheres for 72 hr.

## Cell immunostaining

Cells were cultured in monolayer in laminin-coated coverslips. After reaching 80% confluence, cells were fixed 20 min at room temperature with 4% PFA. Cells were washed three times in cold PBS and incubated 20 min in permeabilization buffer (Triton X-100 0.05%, Goat serum 5%, PBS). Cells were incubated overnight at 4°C with primary antibodies against insulin (Cell Signaling, 1/100), glucagon (Proteintech, 1/100), somatostatin (Peninsula Laboratories International, 1/100), secretogranin 2 (Invitrogen, 1/250) and synaptophysin (Invitrogen, 1/250). GnRH (1/500), VIP, galanin (Abcam, 1/500), oxytocin (Peninsula, 1/500), CRF antibodies (Peninsula, 1/500). Cells were incubated in secondary antibodies (Goat anti-rabbit and goat anti-mouse conjugated with Alexa Fluor 488 and 555, respectively, Invitrogen, 1/500) and after 3 times of wish with cold PBS were mounted (VECTA-SHIELD mounting media with DAPI). In order to visualize the secretory vesicles in higher details, deconvolution analysis was performed based on the published methods in the literature (*Blanco et al., 2011*; *van Steensel et al., 1996*) which allowed to reduce background unspecific signals. Briefly, point spread function (PSF) of each channel (488, 555, 633) was designed according to Leica SP8 confocal microscope parameters (magnification, numerical aperture and media) and image resolution. Merged images were splitted and each channel was de-convolved for five iterations at the respective PSF with 3D de-convolution function in Image J software.

## Sphere and islet immunostaining

Spheroid immunofluorescence: htNSC$^{PGHM}$ Spheres were collected 48 hr after seeding, fixed for 20 min at room temperature with 4% PFA and washed three times in cold PBS with mild centrifugation to avoid disruption of spheroid structure. Free settling (5 min) was employed to separate large-sized spheres in the bottom from small-sized spheres in the supernatant. Spheres were cryostat at 15 μm, then permeabilized and incubated with primary antibodies against pancreatic islet markers (Insulin 1/100, Cell Signaling; Glucagon 1/100, Proteintech; Somatostatin 1/100, Peninsula Laboratories International; Nkx2.2 1/100, EMD Millipore; GLUT2, Novus, 1/100; Kir6.2 1/500, Alomone Labs; PC1/PC3 1/100, EMD Millipore; SOX2 1/100, R and D Systems; Nestin 1/100, EMD Millipore; CD81 1/100, Santa Cruz; TSG101 1/100, Santa Cruz; CCK 1/50, Invitrogen; PACAP 1/100, Peninsula Laboratories International). Neurospheres were incubated with secondary antibody (Goat anti rabbit/ mouse conjugated with Alexa Fluor 488, Invitrogen, 1/500) and mounted (Vecta shield mounting media with DAPI). Adult mouse pancreatic islets were isolated as described before (*Lacy and Kostianovsky, 1967*). Purified mouse islets were fixed 20 min at room temperature with 4% PFA and immunostained following the same procedure and antibodies listed above. All samples were analyzed by con-focal microscopy (Leica SP8 Confocal unit) and post-image analysis were performed with FIJI (Image J). The inner region within 2/3 radius was defined as the center of a sphere or pancreatic islet, while the outer region of 1/3 radius was defined as periphery of these structures.

## Hormone and exosomal miRNA release assays

Hormone release assay htNSCs cells were cultured and let 48 hr to develop spheroids organization. Spheroids were collected and centrifuged at 600 rpm and washed three times to eliminate culture media supplements. The spheroids were resuspended in secretion media (300 μL), consisting in glucose-free Neurobasal media without B27 supplement, BSA 5 mg/mL and stimulated with low or high glucose. Mouse pancreatic islets secretion assay was developed as previously described (*Zúñiga-Hertz et al., 2015*). Secretion supernatant was freshly used for ELISA quantification of Insulin (Alpco, 80-INSMSU-E01, E10) and somatostatin (LSBio LifeSpan BioSciences, Inc, LS-F12622) and by chemiluminescence for Glucagon (EMD Millipore, EZGLU-30K). Exosomal miRNA analysis: Secreted exosomes from cells were purified and analyzed as described previously (*Zhang et al., 2017*). In brief, cells were maintained in exosome-free medium, then exosomes from collected medium were immediately isolated through differential centrifugation. Isolated exosomes were assessed with picoRNA and small RNA chips via Agilent 2100 Bioanalyzer system at the Molecular Pathology Platform, Herbert Irving Comprehensive Cancer Center, Columbia University.

## In vivo sphere implantation

In vivo sphere implantation htNSCs and its derived cells were cultured in NSC medium composed of neurobasal-A, 2% B27 without vitamin A, GlutaMAX supplement, bFGF and EGF. Cells were seeded in Ultra-Low Attachment dishes (Corning) at $5 \times 10^5$ cells·mL$^{-1}$ one day before cell implantation into mice. At the day of implantation, htNSC spheres were collected by low-speed centrifugation (300 rpm for 5 min), and mixed with high-concentration rat tail collagen solution (Corning) and gelled at 37°C for 30 min. The solidified cell-embedded collagen gel was gently compressed to remove liquid and form an implantable collagen sheet as previously described (*Mi et al., 2010*). During collagen gel solidification, a mouse was anesthetized, cleaned, shaved, sterilely prepared for abdominal section. A 0.5 to 1 cm middle section was created at about 0.5 cm below the chest xiphoid to expose the abdominal cavity. The great omentum flap was found and retrieved at the lower rim of the stomach, and was laid flat with occasional PBS moisturization. Cell-containing or empty collagen sheets were then wrapped and secured into the great omentum flap by sutures (*Barthel et al., 2012*). The omentum flap with the collagen sheet was returned to its original location at the end, and the mouse was closed for post-operation maintenance.

## STZ mouse model

C57BL/6J mice were obtained from Jackson Laboratories and group-housed under standard conditions in a temperature- and humidity-controlled facility with 12 hr light:12 hr dark cycles. All mice in this study were kept on a standard normal chow diet obtained from LabDiet (5001, 4.07 kcal·gram$^{-1}$). All procedures of animal studies were approved by the Institutional Care and Use Committee of Albert Einstein College of Medicine (protocol #20171210, #20170812, #20171209, #20171208, #00001111). At the day of streptozotocin (STZ) administration, mice were briefly fasted for 4–6 hr and weighted. STZ powder was dissolved in sterile citrate buffer (pH 4.5) and was immediately i.p. injected into mice at the concentration of 180 mg/kg body weight according to the protocol established in the literature (*Deeds et al., 2011*). Subsequently, 2% sucrose in drinking water was supplied to mice in the first night after STZ administration to counteract hypoglycemia. STZ-induced diabetes in mice was confirmed by blood glucose measurements higher than 350 mg·dL$^{-1}$ in two consecutive days.

## Histology and biochemistry

To analyze morphological changes in pancreatic islets, the pancreas was isolated together with spleen and fixed during 24 hr with PFA 4% at 4°C and subsequently replaced with ethanol absolute. Pancreas was immobilized in paraffin and sectioned. Ten 5µm-sections (each section separated by 5 µm) were mounted and stained with Hematoxylin and Eosin and subsequently scanned with a Perkin Elmer P250 High-Capacity Slide Scanner. Images were visualized and analyzed in Panoramic Viewer software (3DHISTECH Ltd). Islet area was calculated by dividing islet area (µm *Gross, 2000*) by the total area (µm *Gross, 2000*) of the pancreas section analyzed; the mean pancreatic islet area value *per* pancreas was calculated by the mean value of islet area analyzed in each of the ten sections analyzed divided by the total amount of pancreas sections (10 sections). To evaluate cell survival in implants, after mice euthanization the implants were recovered and fixed overnight in PFA 4% at 4°C. The fixation media was discarded and replaced with 30% sucrose and subsequently the implants were placed in Optimal cutting temperature media (Scigen Tissue Plus) and sectioned in cryostat in 10 µm sections. All sections were mounted in microscope slides and imaged for cells of mCherry fluorescence in Leica SP8 Confocal unit 20X magnification. Cell survival was measured as the percentage of GFP- or insulin promoter-selected cells in the total number of cells (by means of DAPI nuclear staining) per implant section. Serum samples obtained from mice under indicated conditions were measured for insulin using an ultrasensitive ELISA kit (ALPCO, 80-INSMSU-E01) according to the manufacturer's instruction.

## Behavioral tests

Animal survival was carefully and closely monitored on daily basis through checking dealth as well as terminal conditions according to the approved protocol. All behavior tests were performed in the behavioral testing room as previously described (*Zhang et al., 2013*). An Anymaze video-tracking system (Stoelting) equipped with a digital camera connected to a computer was used for the whole

course of animal activities in training and experimental sessions of behavioral tests, including the following tests. (1) Grip test: Mouse was lifted by the tail and placed on a homemade square grid (1 cm mesh size), the grid was then inverted 30 cm over a soft pad, and the mouse was allowed to hang by paws for 2 to 10 min according to experimental conditions. The time that the mouse was able to hang was recorded during the test period. Three repeats were performed for each mouse with at least 10 min rest between each trial. (2) Rotarod: Mouse was trained on a rotarod (Columbus instruments) that was moving at a speed (4 to 6 r.p.m.) for a period of 1 to 2 min. After a 10 min rest, each mouse was given three trials, during which the rotarod started at 6 r.p.m. and accelerated by 2 to 3 r.p.m. per min until 10 min, and there was 30 min rest period between each trial. (3) Treadmill: Treadmill test was performed using a Treadmill Simplex II (Columbus Instruments), which has a capacity of exercising up to six mice simultaneously in individual lanes. Mouse was warmed up for protection against running injury and failure before experimental running. For acclimatization to the treadmill, mouse was placed on the treadmill running at a low speed (6–8 meter per min) for 2 to 5 min. The test session started at a speed of 8 meter per min, and the speed was increased by two meter per min every min for 10 min. The time at which the mouse retired was recorded. (4) Open field locomotion: Open field test was performed as described previously (*Seibenhener and Wooten, 2015*). Briefly, a white plastic chamber which measured 40 cm (length) x 40 cm (width) x 48 cm (height) was wiped with 75% ethanol before each trial. Then the mouse was put in the lower right corner of the chamber, facing to the wall. The route of the mouse in the chamber was monitored for 10 min.

## Statistics

All measured data were presented as mean ± SEM. Sample sizes with sufficient power were designed according to our published studies, relevant literature and preliminary studies. ANOVA and appropriate (such as Tukey) post-hoc tests were used for comparisons for experiments of involving more than two groups. Student's t-test was used when studies involved only two groups for comparisons. Survival curves were analyzed using Log-rank (Mantel-Cox) survival test. Linear regression was performed when appropriate to analyze the relationships between survival and metabolic parameters. All key experiments were repeated at least twice independently. Data analyses did not encounter any outlier or other exclusions. Whenever necessary, experimental performers were blind to group information before data were obtained. Software for performing statistics included Excel and GraphPad Prism, and p value of less than 0.05 was considered statistically significant.

## Acknowledgements

The authors sincerely thank Cai laboratory personnel and Einstein Pathology Core for providing technical supports. This study is supported NIH AG031774, DK099136, HL147477 and DK121435 (all to D Cai).

## Additional information

### Funding

| Funder | Grant reference number | Author |
|---|---|---|
| National Institute on Aging | AG031774 | Dongsheng Cai |
| National Institute of Diabetes and Digestive and Kidney Diseases | DK099136 | Dongsheng Cai |
| National Heart, Lung, and Blood Institute | HL147477 | Dongsheng Cai |
| National Institute of Diabetes and Digestive and Kidney Diseases | DK121435 | Dongsheng Cai |

The funders had no role in study design, data collection and interpretation, or the decision to submit the work for publication.

## Author contributions

Yizhe Tang, Data curation, Formal analysis (statistics), Investigation (performance), Methodology (cloning, cell line generation, biochemistry, surgery, phenotyping), Participating in interpretation, Writing assistance; Juan Pablo Zuniga-Hertz, Data curation, Formal analysis (statistics), Investigation (performance), Methodology (cell culture, vesicle assay, biochemistry, staining, histology), Participating in interpretation, Writing assistance; Cheng Han, Data curation, Formal analysis (statistics), Investigation (performance), Methodology (cell culture, surgery, phenotyping), Participating in interpretation; Bin Yu, Data curation, Formal analysis (statistics), Investigation (performance), Methodology (cell culture, staining, animal behaviors), Participating in interpretation, Writing assistance; Dongsheng Cai, Conceptualization, Project design (aims, structure, strategies, experiments), Resources, Supervision, Funding, Methodology (strategies, approach designs, analysis), Investigation (evaluation, development, quality control), Data curation, Formal analysis (evaluation, approval), Validation, Interpretation (leading role), Visualization, Project administration, Writing (original writing, editing, revision, finalizing)

## Author ORCIDs

Dongsheng Cai ⬦ https://orcid.org/0000-0002-4187-1019

## Ethics

Animal experimentation: All animal procedures in this study were approved by the IACUC of Albert Einstein College of Medicine. (protocol #20171210, #20170812, #20171209, #20171208, #00001111).

## Decision letter and Author response

Decision letter https://doi.org/10.7554/eLife.52580.sa1
Author response https://doi.org/10.7554/eLife.52580.sa2

# Additional files

## Supplementary files

• Transparent reporting form

## Data availability

All data generated or analysed during this study are included in the manuscript and supporting files.

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
