## [Decision Letter]

**Acceptance summary:**

This provocative and innovative study provides evidence that hypothalamic neural stem cells secrete various peptides (including insulin, *Sst* and more), and are capable to improve general health and survival in different unrelated diseases. The revised manuscript has met all outstanding concerns. This study of these interesting and apparently versatile cells will be of interest to many in the stem cell field.

**Decision letter after peer review:**

Thank you for sending your article entitled "Multifaceted secretion of hypothalamic spheres and therapy against multiple fatal diseases via peripheral application" for peer review at *eLife*. Your article has been evaluated by two peer reviewers, and the evaluation has been overseen by a Reviewing Editor and Marianne Bronner as the Senior Editor.

Summary:

This provocative and innovative study provides evidence that hypothalamic neural stem cells secrete various peptides (including insulin, *Sst* and more), and are capable to improve general health and survival in different unrelated diseases.

Essential revisions:

While both reviewers agree that your findings are very interesting, reviewer 2 raises some important questions about reproducibility and the consistency and quality of some of the data that must be addressed. It is not really clear whether these key issues can be resolved in a reasonable timeframe, but we would anticipate that a revision would normally be completed within two months.

1) One major concern is how consistent the 3D spheres are in the expression of peptides and exosome profiles. Since these 3D spheres were obtained through many different steps involving various vector expression and isolation, it is conceivable that there would be some random distributions in cell types expressing various peptides etc. From the available pictures, it appears that these 3D spheres are present in variable size. More thorough analysis on these 3D spheres to have a more complete picture on these 3D spheres as a whole is required.

2) There are inconsistencies in a lot of data presented. Figure 3E showed that STZ sham mice had 50% survival at day 40; however, in Figure 3C, data points at day 45 appears to be a few mice less than the starting point. The same is true for Figure 4A and 4C; and for Figure 5A and C.

3) Figure 4C showed that the STZ sham had a very low survival rate, most died with 6 days. If this is the case, how could the study be conducted? Also the survival rate is dramatically different from that shown in Figure 3E. These inconsistencies point to poor quality of the data presented.

4) Figure 1G showed that the 3D spheres had a 100 fold less capacity in releasing insulin compared to control islets. How can this low capacity in releasing insulin explain the significantly increase in insulin levels presented in Figure 3D?

5) Some key data presented were not adequately explained. For example, what is "Disease activity index" in Figure 5B? According to Figure 5C, close to 50% mice died after day 15, how the index remains stable from day 10 and beyond in Figure 5B? In Figure 6B, what is EAE "clinical score"? Again according to Figure 6K, 50% mice died at day 20 and beyond; however, the clinical score remained stable across that period. Is death an extreme mark of clinical score?

6) Data presented in Figures 5 and 6 are weak and superficial, lacking insights on underlying mechanism, and therefore are premature.

7) Given the perceived variation in 3D spheres, a detailed analysis in study subjects to correlate fasting glucose (Figure 3C), fasting insulin (Figure 3D) and animal survival (Figure 3E), if these studies used the same cohort of mice, would add more weights to the role of insulin secretion from 3D spheres in glucose and healthy state.

[Editors' note: further revisions were suggested prior to acceptance, as described below.]

Thank you for resubmitting your work entitled "Multifaceted secretion of hypothalamic spheres provides pro-survival effects from peripheral implantation" for further consideration by *eLife*. Your revised article has been reviewed by two peer reviewers and the evaluation has been overseen by Marianne Bronner as the Senior Editor, and a Reviewing Editor.

Both reviewers found that your revision addressed all major points and strengthened the conclusion of the study. You may wish to give some consideration to the one remaining point raised by the first reviewer prior to acceptance, as outlined below:

1) This updated version has largely addressed the concerns raised in my previous review. Especially, new data were added to increase the level of confidence on the conclusion drawn about the effects on STZ-induced diabetic models, and new ways of data presentation have cleared the confusion in the previous version. However, the whole Supplementary Figure 7 should be eliminated as the evidence to support the beneficial effects in those two disease models is still lacking, and this Supplementary Figure 7 seems not to be logically relevant to the rest of the data presented. Importantly, the data and associated finding on the 3D spheres and effects on STZ-induced type 1 diabetes still remain significant without Figure 7.

---

## [Author Response]

Essential revisions:While both reviewers agree that your findings are very interesting, reviewer 2 raises some important questions about reproducibility and the consistency and quality of some of the data that must be addressed. It is not really clear whether these key issues can be resolved in a reasonable timeframe, but we would anticipate that a revision would normally be completed within two months.1) One major concern is how consistent the 3D spheres are in the expression of peptides and exosome profiles. Since these 3D spheres were obtained through many different steps involving various vector expression and isolation, it is conceivable that there would be some random distributions in cell types expressing various peptides etc. From the available pictures, it appears that these 3D spheres are present in variable size. More thorough analysis on these 3D spheres to have a more complete picture on these 3D spheres as a whole is required.

We want to explain that 3D spheres are not simply a mixture of cells but a biological process involving cell type organization; thus, some cell subtypes could be characteristically (rather than randomly) distributed in the sphere.

To adequately address this issue, we re-performed experiments and characterized the distribution patterns of each cell subtype in the htNSC spheres (Figure 3A, B for large-sized and Figure 3—figure supplement 1A, B for small-sized) compared to pancreatic islets (Figure 3—figure supplement 2A, B).

To summarize, insulin-positive cells were distributed both in the center and periphery, only slightly differing from pancreatic islets which are known to contain insulin-positive cells (i.e., β cells) in the center slightly more than periphery. While glucagon- and somatostatin-positive cells (known as α and δ cells) are distributed typically in the periphery of pancreatic islets (Figure 3—figure supplement 2A-B), it was the case for somatostatin but not for glucagon in htNSC spheres (Figure 3A-B, Figure 3—figure supplement 1A-B). Pancreatic islet biomarkers PC1/3, GLUT2, KIR6.2 and NKX2.2 were widely expressed in htNSC spheres (Figure 3A-B, Figure 3—figure supplement 1A-B) in manners which were similar to pancreatic islets (Figure 3—figure supplement 2). Unlike pancreatic islets which barely expressed NSC markers *Sox2* and nestin, both were strongly and universally expressed in htNSC spheres (Figure 3A-B, Figure 3—figure supplement 1–2). Regarding exosomes, we examined two exosomal biomarkers CD81 and TSG101, clearly showing that they were widely expressed in htNSC spheres but barely in pancreatic islets (Figure 4A-B). Altogether, all these data again indicate that these htNSC structures contained the mixed features of pancreatic islets and neurospheres.

Regarding neurosphere sizes, they are by nature variable (which is a general feature of these spheres). Since the reviewer mentioned this, we analyzed large-sized vs. small-sized spheres and in the same manner and presented large sizes in Figure 3A-B and small sizes in Figure 3— figure supplement 1A-B. These results confirmed that large and small spheres have the similar structure of cell subtypes.

2) There are inconsistencies in a lot of data presented. Figure 3E showed that STZ sham mice had 50% survival at day 40; however, in Figure 3C, data points at day 45 appears to be a few mice less than the starting point. The same is true for Figure 4A and 4C; and for Figure 5A and 5C.

Animal numbers between previous Figure 3C and 3E are indeed consistent (current Figure 5D and E). This misunderstanding was due to the overcrowded font of previous Figure 3C: because blood glucose levels of mice at the initial time points were very similar, as we plotted them according to the large scale of Y-axis, the dots of representing individual mice mostly overlapped. Now, we changed the font by using small-sized dots (currently Figure 5D) which helps although there are still overlapping points. To help solve this issue, we have now provided n number for each time point in its legend (Figure 5D) and also include these details in Author response table 1. Previous Figure 3E (current Figure 6A) involved too many time points to be detailed in its legend, so we include these details in Author response table 1.

**Author response table 1. resptable1:** 

Figure 5D: animal number of each group and each time point		Figure 6A: animal number of each group and each time point (Please note that these cohorts were different from Figure 5D)
Day	+STZ, Sham	+STZ, htNSC^PGHM^		Day	+STZ, Sham	+STZ, htNSC^PGHM^
-5	30	39		-5	26	25
5	30	39		5	26	25
9	29	39		7	25	25
16	26	37		9	25	25
22	22	36		14	23	25
30	15	34		16	23	25
36	14	32		21	22	23
45	12	29		22	22	23
				25	22	21
				26	21	21
				28	18	21
				30	17	21
				32	16	21
				36	16	21
				42	13	21
				45	12	20
				52	10	20
				55	9	19
				56	8	19
				57	8	18
				65	7	17
				77	5	15

Animal numbers between previous Figure 4A and C are consistent (current Figure 7A and C). Also because these figures involved too many time points and many groups to be detailed in their legends, we provide these details in Author response table 2.

**Author response table 2. resptable2:** 

Figure 7A: animal number of each group and each time point		Figure 7C: animal number of each group and each time point
Week	shIPC	shIns	shCtrl	Sham	shRab		Day	shIPC	shIns	shCtrl	Sham	shRab
1	20	22	25	15	15		0	21	22	25	15	15
2	20	22	24	12	12		3	20	22	25	15	15
3	17	19	24	9	10		7	20	22	25	15	15
4	15	16	23	7	8		12	20	22	24	14	14
5	13	15	22	5	6		14	20	22	24	12	12
6	11	11	21	3	4		15	19	22	24	12	10
							16	18	21	24	12	10
							18	18	20	24	11	10
							19	18	20	24	10	10
							21	17	19	24	9	10
							22	16	19	24	9	9
							25	16	19	24	8	9
							26	15	18	24	7	8
							27	15	17	23	7	8
							28	15	16	23	7	8
							29	14	16	23	6	8
							30	14	16	22	6	8
							31	14	16	22	5	8
							33	13	15	22	5	8
							34	13	15	22	5	7
							35	13	15	22	5	6
							36	13	14	21	5	5
							38	13	13	21	5	5
							39	13	12	21	4	5
							42	11	11	21	3	4
							43	9	10	20	3	4
							46	9	10	19	3	4
							48	9	10	18	3	4
							49	9	9	18	3	4
							52	8	9	17	3	4

Animal numbers between previous Figure 5A and 5C are also consistent (currently Figure 7— figure supplement 1B and D). Again because these figures involved too many time points to be detailed in their legends, we include these details in Author response table 3.

**Author response table 3. resptable3:** 

Figure 7—figure supplement 1B: Animal number of each group and each time point		Figure 7—figure supplement 1D: Animal number of each group and each time point
Day	Veh+Sham	DSS+Sham	DSS+ htNSC^PGHM^		Day	Veh+Sham	DSS+Sham	DSS+ htNSC^PGHM^
0	10	9	9		0	10	9	9
1	10	9	9		1	10	9	9
2	10	9	9		2	10	9	9
3	10	9	9		3	10	9	9
4	10	9	9		4	10	9	9
5	10	9	9		5	10	9	9
6	10	9	9		6	10	9	9
7	10	9	9		7	10	9	9
8	10	9	9		8	10	9	9
9	10	9	9		9	10	9	9
10	10	9	9		10	10	9	9
11	10	9	9		11	10	9	9
12	10	9	9		12	10	9	9
13	10	8	9		13	10	8	9
14	10	6	9		14	10	6	9
15	10	6	9		15	10	6	9
16	10	6	9		16	10	6	9
17	10	6	9		17	10	6	9
18	10	5	9		18	10	5	9
19	10	5	9		19	10	5	9
20	10	5	9		20	10	5	9
21	10	5	9		21	10	5	9
22	10	5	9		22	10	5	9
23	10	5	9		23	10	5	9
24	10	5	9		24	10	5	9
25	10	5	9		25	10	5	9
26	10	5	9		26	10	5	9
27	10	5	9		27	10	4	9
					28	10	4	9
					29	10	4	9
					30	10	4	9

3) Figure 4C showed that the STZ sham had a very low survival rate, most died with 6 days. If this is the case, how could the study be conducted? Also the survival rate is dramatically different from that shown in Figure 3E. These inconsistencies point to poor quality of the data presented.

Most likely because the text font in X-axis of this figure was too small (apologize for this inconvenience), the reviewer misread this number. Our previous Figure 4C (current Figure 7C) shows that most mice died 60 days, not 6 days. We have now improved the font of this figure.

4) Figure 1G showed that the 3D spheres had a 100 fold less capacity in releasing insulin compared to control islets. How can this low capacity in releasing insulin explain the significantly increase in insulin levels presented in Figure 3D?

This issue is due to our inadequate writing. Although 3D spheres were 100-fold less capable of releasing insulin compared to islets, we implanted 10^7^ htNSC cells (which were around 10^4^ to 10^5^ spheres), while a mouse has only about 300 pancreatic islets. So the quantity advantage of spheres helped. We have now added this discussion in the revised manuscript.

5) Some key data presented were not adequately explained. For example, what is "Disease activity index" in Figure 5B? According to Figure 5C, close to 50% mice died after day 15, how the index remains stable from day 10 and beyond in Figure 5B? In Figure 6B, what is EAE "clinical score"? Again according to Figure 6K, 50% mice died at day 20 and beyond; however, the clinical score remained stable across that period. Is death an extreme mark of clinical score?

“Disease activity index” was based on surviving mice (currently moved to Figure 7—figure supplement 1C). This analysis has been well established in the literature (for example, PMID10759763), which cannot assess the involved physiological conditions of a dead mouse. Also, as well published, this pathological index is stable for a few weeks after the disease is established. “EAE clinical score” did include death as the worst score and we re-analyzed the data through adopting a published 0–6 scoring system which better reflects the contribution of death, as detailed in the Materials and methods (for example, PMID12023381). In our EAE without sphere treatment (current Figure 7—figure supplement 3B), this EAE clinical score peaked around Day 24 which agreed with the survival status (current Figure 7—figure supplement 3G), and afterwards these mice stopped to die while their health became improved and recovered, so our curve pattern agrees with these conditions.

6) Data presented in Figures 5 and 6 are weak and superficial, lacking insights on underlying mechanism, and therefore are premature.

We did consider if we wanted to include these two unrelated disease models, but in the end, we think it should be helpful to include them, at least because they can provide additional information in the context of the interest in this paper. Given the reviewer’s comment, we agree that these two models probably should be presented as supplementary figures, and thus we have moved them to Figure 7—figure supplements 1-4.

7) Given the perceived variation in 3D spheres, a detailed analysis in study subjects to correlate fasting glucose (Figure 3C), fasting insulin (Figure 3D) and animal survival (Figure 3E), if these studies used the same cohort of mice, would add more weights to the role of insulin secretion from 3D spheres in glucose and healthy state.

Thanks for this insightful suggestion. Hence, we performed correlation analyses between blood glucose/insulin levels and survival in the STZ model. These analyses are shown in Figure 6C, D, E and F. As shown, there was a lack of correlation between hyperglycemia and survival in STZ model, suggesting that death was not simply a result of such hyperglycemia when blood glucose was high enough. The sphere-treated STZ mice had a modest reduction in hyperglycemia (they were still diabetic) and a much greater improvement of survival; our analysis revealed only a tendency of correlation between blood glucose and survival, indicating that this modest benefit of dropping blood glucose did not substantially contribute to the survival. In terms of blood inulin, there was also a lack of correlation between its level and the survival in STZ mice. In sphere-treated STZ mice, the correlation between insulin and survival just marginally become significant, suggesting that elevation in blood insulin can partially contribute to the improved survival of these diabetic mice.

[Editors' note: further revisions were suggested prior to acceptance, as described below.]

Both reviewers found that your revision addressed all major points and strengthened the conclusion of the study. You may wish to give some consideration to the one remaining point raised by the first reviewer prior to acceptance, as outlined below:1) This updated version has largely addressed the concerns raised in my previous review. Especially, new data were added to increase the level of confidence on the conclusion drawn about the effects on STZ-induced diabetic models, and new ways of data presentation have cleared the confusion in the previous version. However, the whole Supplementary Figure 7 should be eliminated as the evidence to support the beneficial effects in those two disease models is still lacking, and this Supplementary Figure 7 seems not to be logically relevant to the rest of the data presented. Importantly, the data and associated finding on the 3D spheres and effects on STZ-induced type 1 diabetes still remain significant without Figure 7.

Thanks to the reviewers for this suggestion. We thus have removed the whole Supplementary Figure 7, and accordingly removed/edited the relevant text.